# Factors associated with violent offenders with mental illness in forensic psychiatric evaluations

Chia-Heng Lin[1,2], Wen-Ching Hsieh[3], Li-Ting Lin[4,5], Chia-Hsiang Chan[4,5,6]*

1 Department of General Psychiatry, Bali Psychiatric Center, Ministry of Health and Welfare, New Taipei City, Taiwan, 2 Institute of Crime Prevention and Corrections, Central Police University, Taoyuan, Taiwan, 3 Department of Psychology, Taoyuan Psychiatric Center, Ministry of Health and Welfare, Taoyuan, Taiwan, 4 Department of General Psychiatry, Taoyuan Psychiatric Center, Taoyuan, Taiwan, 5 Institute of Brain Science, National Yang Ming Chiao Tung University, Taipei, Taiwan, 6 Department of Psychology, Chung Yuan Christian University, Chungli, Taiwan

* cscott1125@gmail.com

## Abstract

### Purpose

The overall crime rate among individuals with severe mental illnesses is similar to that of the general population, although some studies suggest a higher risk of violent crime among this group. Empirical research on factors associated with violent crime among individuals with mental illnesses in East Asia remains limited.

### Methods

This study examined 648 offenders referred for forensic psychiatric evaluation by the criminal justice system to explore the relationship between severe mental illness, substance-related and addictive disorders, and violent crime. Demographic, clinical, forensic, and both static and dynamic factors were analyzed using bivariate analysis and multivariate logistic regression. We also tested the moderating effects of gender, history of violent crime, poor treatment adherence, and comorbid substance-related and addictive disorders on the association between severe mental illness and violent crime.

### Results

The results showed that violent offenders were more likely to be male and to have never undergone a psychiatric evaluation prior to the offense, compared to non-violent offenders. Severe mental illness, substance-related and addictive disorders, single status, unemployment, and poor treatment adherence were not significantly associated with violent crime. Furthermore, gender, poor treatment adherence, a history of violent crime, and comorbid substance-related and addictive disorders did not significantly moderate the relationship between severe mental illness and violent crime.

**Data availability statement:** All relevant data are available within the paper and its Supporting Information files. We have also uploaded the de-identified data onto the site The Qualitative Data Repository https://qdr.syr.edu/ The DOI is https://doi.org/10.5064/F6JZOYI6. Please also refer to the site https://data.qdr.syr.edu/dataset.xhtml?persistentId=doi%3A10.5064/F6JZOYI6.

**Funding:** The author(s) received no specific funding for this work.

**Competing interests:** The authors have declared that no competing interests exist.

## Conclusion

These findings emphasize that individuals with severe mental illness should not be automatically linked to violent offending. A comprehensive evaluation of offenders with severe mental illness is crucial, alongside a deeper understanding of their treatment and reintegration needs.

## 1. Introduction

At the start of the 21st century, the World Health Organization (WHO) highlighted the critical role of mental health. Approximately 25% of the global population will experience a mental illness at some point in their lives. However, stigma remains a significant barrier, often preventing affected individuals from receiving proper treatment, obtaining stable housing, and securing employment [1].

Film and television frequently depict individuals with mental illness as irrational, dangerous, or fundamentally flawed. The media also disproportionately reports on crimes committed by those with severe mental illness, further shaping public perceptions [2,3]. Common stereotypes include the belief that mental illness is a personal choice, that affected individuals are undeserving of sympathy, lack self-control, or are incapable of making independent decisions [4,5]. These misconceptions fuel stigma and perpetuate harmful narratives about mental illness.

Assessing the risk factors for criminal behavior in individuals with mental illness requires consideration of not only active psychiatric symptoms [6] but also substance use, socioeconomic status, and environmental factors [7]. Furthermore, when evaluating the risk of violent crime in this population [8], factors such as gender, history of violent crime, illicit substance and alcohol use, treatment adherence, and psychiatric diagnosis type must also be considered.

Fazel and his research team conducted a meta-analysis of studies from English-speaking countries, using schizophrenia as a case study [9]. Their findings indicate that individuals with schizophrenia have an odds ratio (OR) of 4.0 for violent offenses compared to the general population under a random-effects model (95% confidence interval [CI] = 3.0–5.3), with high heterogeneity ($I^2 = 88\%$, 95% CI = 78–91%). Even after adjusting for socioeconomic factors, the OR remained at 3.8 (95% CI = 2.6–5.0), although heterogeneity remained high ($I^2 = 84\%$, 95% CI = 74–90%). The violent crimes analyzed in this study included homicide, attempted homicide, aggravated assault, sexual offenses, robbery, intimidation, unlawful detention, battery, arson, and domestic violence. However, in Western countries, the likelihood of an individual with schizophrenia committing homicide is approximately 0.01% annually, while the probability of any violent crime is approximately 0.6% per year [10]. Some clinical psychiatrists argue that individuals with schizophrenia are more often involved in legally defined minor assaultive behaviors than in severe violent crimes [11,12]. These behaviors are frequently linked to situational disputes, personality traits, or substance use, with treatment adherence also playing a crucial role [11,12].

A study in Sweden examining medical and criminal records [13] investigated the link between bipolar disorder and violent crime. Compared to the general population, individuals with bipolar disorder had an adjusted odds ratio (AOR) of 2.3 for violent crimes (95% CI = 2.0–2.6). The risk was significantly higher among those with comorbid substance use disorders, with an AOR of 6.4 (95% CI = 5.1–8.1). Conversely, individuals with bipolar and related disorders without comorbid substance use had a significantly lower AOR of 1.3 (95% CI = 1.0–1.5). Notably, within the broader category of bipolar and related disorders—including mania, depression, and psychotic features—there were no significant differences in violent crime risk among these subgroups. However, variations in how violent crime was defined and differences in research methodologies across studies should be considered when interpreting these findings.

The global prevalence of substance use disorders is approximately 2.2% [14], encompassing alcohol, stimulants, sedatives, hypnotics, cannabis, opioids, and hallucinogens. Individuals who use these substances have a 4- to 10-fold higher risk of committing violent crimes than non-users [15], a risk influenced by the complex neurophysiological changes induced by substance use [16]. For instance, both long-term alcohol consumption [17] and short-term binge drinking [18] can disrupt serotonin balance and overstimulate dopamine, leading to heightened mental activity, increased sensory-seeking behavior, provocation, and impulsivity [19,20]. Stimulants, meanwhile, enhance dopamine, serotonin, and norepinephrine release in the central nervous system, resulting in euphoria, heightened sensory perception, increased libido, anxiety, confusion, agitation, hallucinations, and delusions. In severe cases, these effects can contribute to violent behavior or suicidal tendencies [21,22]. However, substance users may also engage in criminal activity for reasons beyond neurophysiological effects, such as financial motives to sustain substance use [23].

Recent large-scale research on this topic in East Asia includes a study by Kim et al. that analyzed judicial databases [24]. Their findings suggest that individuals with severe mental illness have a higher likelihood of committing homicide, arson, and substance-related offenses than the general population. However, previous studies on individuals with mental illness reintegrated into society after hospital discharge suggest that when symptoms are stable, their risk of posing a threat is comparable to that of the general population. Conversely, individuals with substance use disorders present a significantly higher risk [25]. Overall, the relationship between substance use disorders, severe mental illness (such as schizophrenia and bipolar disorder) [26], and violent crime is unlikely to be a simple additive effect. Furthermore, variations in offense classification and sample sources may significantly influence analytical outcomes.

Data on offenders with mental illness can be sourced from medical records, social welfare systems, the criminal justice system, and forensic psychiatric evaluations. These evaluations are typically conducted by psychiatric experts, psychologists, and social workers who compile comprehensive reports. By integrating both dynamic and static risk factors associated with criminal behavior, these reports serve as valuable tools for analysis and practical application. We believe that through these data, we can identify variables potentially relevant to the relationship between mental illness, substance use, and criminal behavior. These variables warrant further investigation and may provide an opportunity to develop an integrated theoretical framework in this field.

Forensic psychiatric evaluations have been instrumental in analyzing crime patterns and identifying risk factors across different regions. In Sweden, forensic psychiatrists have used these reports to analyze crime patterns among individuals with intellectual disabilities [27]. In Lithuania, experts have examined them to identify potential risk factors for homicides committed under the influence of alcohol [28]. Similarly, researchers in Egypt have extracted variables from forensic psychiatric reports in both criminal and civil cases to assess potential criminogenic factors among offenders [29]. These studies highlight the value of forensic psychiatric evaluations in providing deeper insights into offenders' medical conditions, treatment needs, correctional strategies, and preventive measures, both before and after trial.

Building on this framework, we aimed to analyze offenders documented in forensic psychiatric evaluation reports to test the following hypotheses: (1) Severe mental illness, including schizophrenia and bipolar disorder, as well as substance-related and addictive disorders, are each associated with violent crime. (2) Gender, treatment adherence, history of violent crime, and comorbid substance-related and addictive disorders influence the relationship between severe mental illness and violent crime.

## 2. Materials and methods

### 2.1. Case demographic information

This retrospective study was conducted at a psychiatric specialty hospital in northern Taiwan, where the population density is approximately 1,800 people per square kilometer and the crime rate is approximately 1,000 per 100,000 people [30]. The forensic psychiatry evaluation team of the hospital conducts forensic psychiatric evaluations for criminal cases referred by prosecutors or judges. These evaluations determine psychiatric diagnoses, criminal responsibility, trial competence, and other forensic considerations, culminating in formal forensic reports. Offenders are typically referred for forensic psychiatric evaluation several months after committing a crime, often having already received psychiatric treatment elsewhere. This study reviewed forensic psychiatric evaluation reports completed at the hospital between October 2009 and February 2022. Civil cases and victim assessments were excluded, yielding a final sample of 648 cases. Diagnoses were based on criteria from either the DSM-IV-TR [31] or the DSM-5 [32]. The evaluation process involved a multidisciplinary team comprising two psychiatrists, one psychologist, and one social worker, all of whom conducted interviews. This study was approved by the ethics committee of the hospital (IRB approval number: B20210722-2). We had no direct contact with the individuals evaluated, and the study did not interfere with their evaluations, treatment, or management. The research team accessed the forensic psychiatric evaluation data on April 24, 2022, for analysis purposes.

### 2.2. Classification of violent crime

The primary dependent variable in this study was violent crime, defined as offenses, including homicide, attempted homicide, aggravated assault, battery, arson, domestic violence, sexual offenses, robbery, intimidation, and unlawful detention [33,34]. Offenders were categorized as "violent" or "non-violent" based on the charges they faced at prosecution, aligning with the established definition of violent crime.

### 2.3. Sociodemographic factors

In this study, we examined key demographic variables, including age, gender, education level, employment status, and marital status. Education level is categorized based on whether an individual completed a compulsory junior high school education. Employment status is classified according to whether an individual was engaged in paid work at the time of the offense. Marital status is divided into three categories—single, married, or separated.

### 2.4. Clinical factors

The clinical factors included in this study are family history of mental illness, substance exposure history, suicide history, prior psychiatric evaluations, and treatment adherence. A family history of mental illness is defined as having a first-, second-, or third-degree relative diagnosed with a mental illness, as documented in the forensic evaluation report. This includes schizophrenia spectrum and other psychotic disorders, bipolar and related disorders, depressive disorders, substance-related and addictive disorders, and intellectual disabilities. Suicide history refers to documented suicide attempts involving actual self-harm, excluding suicidal ideation. The absence of prior psychiatric evaluation is defined as having never undergone a psychiatric evaluation before the offense based on forensic evaluation records. Poor treatment adherence is classified as following prescribed pharmacological or non-pharmacological treatment at a rate of less than 80% in terms of the number of sessions attended, frequency, or timing, as recorded in medical reports [35].

### 2.5. Forensic factors

Forensic factors assessed in this study include a history of violent crime and diagnoses from forensic psychiatric evaluations. A history of violent crime is defined as any previously documented offenses in forensic psychiatric evaluation reports, including homicide, attempted homicide, aggravated assault, sexual offenses, robbery, intimidation, unlawful

detention, battery, arson, and domestic violence. Diagnoses from forensic psychiatric evaluations are classified according to the diagnoses recorded in these reports. The primary diagnostic categories include schizophrenia spectrum and other psychotic disorders, bipolar and related disorders, depressive disorders, substance-related and addictive disorders, neurocognitive disorders, personality disorders, and intellectual disabilities. For analytical purposes, schizophrenia spectrum and other psychotic disorders, along with bipolar and related disorders, are grouped under the severe mental illness category [26].

### 2.6. Static and dynamic factors of violent crime

Our study defines static and dynamic factors of violent crime based on demographic, clinical, and forensic evaluation data from forensic psychiatric evaluation reports, as well as meta-analyses of recidivism risk factors among offenders with mental illnesses [8]. Static factors include age, male gender, incomplete compulsory education, a family history of severe mental illness, a prior history of violent crime, and the absence of prior psychiatric evaluation before the offense. Dynamic factors include unemployment, single marital status, severe mental illness, substance-related and addictive disorders, and poor treatment adherence.

### 2.7. Statistical analyses

This study includes both continuous and categorical variables across demographic, clinical, and forensic evaluation-related factors. Continuous variables were reported as means and standard deviations, while categorical variables were expressed as frequencies and percentages. Bivariate analyses were conducted using independent sample $t$-tests for continuous variables and chi-square tests for categorical variables. Multiple logistic regression was performed to examine associations between static and dynamic crime factors (independent variables) and offender classification (violent vs. non-violent) as the dependent variable. Independent variables were entered into the statistical model using the Enter method. Model fit in logistic regression analysis were assessed using the Hosmer–Lemeshow goodness-of-fit test. Statistical significance was set at $p < 0.05$, with all tests conducted as two-tailed analyses.

To assess whether the static factors—male gender and history of violent crime—and the dynamic factors—poor treatment adherence and diagnoses from forensic psychiatric evaluation of substance-related and addictive disorders—moderate the relationship between severe mental illness and violent crime, a moderation analysis was conducted using Hayes' PROCESS macro in SPSS [36]. Each factor was examined individually, with other factors included as covariates to control for potential confounding effects. Multicollinearity diagnostics indicated acceptable levels across all predictors, with variance inflation factors (VIFs) below 2.0, suggesting that multicollinearity was not a concern in the regression models.

In addition, to minimize the potential influence of crime classification on the study results, we conducted the same set of comparisons and analyses using the most violent and the most common non-violent crime categories. Specifically, we compared homicide and theft offenders in terms of sociodemographic, clinical, and forensic factors, followed by regression analyses of static and dynamic factors, and finally, moderation analyses. This approach was intended to determine whether crime classification had a substantial impact on the study findings.

## 3. Results

### 3.1. Sociodemographic factors

The sample had a mean age of $39.20 \pm 11.98$ years, with 79.0% male, 18.7% not completing compulsory education, 90.1% unemployed, and 90.9% single. Table 1 presents the differences in the demographic factors of violent and non-violent offenders. Violent offenders were significantly more likely to be male (89.8% vs. 69.5%, $p < 0.001$), slightly less likely to be unemployed (87.5% vs. 92.4%, $p = 0.047$), and had a lower proportion of single status (87.2% vs. 94.2%, $p = 0.002$). Additionally, a higher proportion of violent offenders were married compared to non-violent offenders (12.2% vs. 5.5%, $p = 0.003$).

**Table 1. Comparison of sociodemographic factors between violent and non-violent offenders.**

| | Violent offenders (n = 304) | | Non-violent offenders (n = 344) | | | |
|---|---|---|---|---|---|---|
| | Mean | SD | Mean | SD | t | p value |
| Age | 38.62 | 12.11 | 39.72 | 11.85 | −1.170 | 0.243 |
| | n | % | n | % | $\chi^2$ | p value |
| Male gender | 273 | 89.8 | 239 | 69.5 | 40.207 | <0.001 |
| Incomplete national education | 60 | 19.7 | 61 | 17.7 | 0.427 | 0.545 |
| Unemployed | 266 | 87.5 | 318 | 92.4 | 4.428 | 0.047 |
| Marital status | | | | | | |
| Single | 265 | 87.2 | 324 | 94.2 | 9.596 | 0.002 |
| Separated | 2 | 0.7 | 1 | 0.3 | 0.472 | 0.603 |
| Married | 37 | 12.2 | 19 | 5.5 | 9.033 | 0.003 |

## 3.2. Clinical factors

In the total sample, 13.6% had a family history of severe mental illness, while 17.3% had never undergone psychiatric evaluation before committing the crime. Moreover, 92.4% exhibited poor treatment adherence. Table 2 presents the clinical differences between violent and non-violent offenders. Violent offenders were significantly more likely to have no prior psychiatric evaluation before the offense than non-violent offenders (24.7% vs. 10.8%, $p < 0.001$).

## 3.3. Forensic factors

In the total sample, homicide cases constituted 5.4%, attempted homicide 6.8%, and sexual offenses constituted 14.5%. Non-violent crimes accounted for 62.7% of the cases, with theft representing 30.6%. Among individuals who underwent forensic psychiatric evaluations, 44.9% were diagnosed with severe mental illness, while 28.4% had substance-related and addictive disorders. Table 3 presents the differences in forensic psychiatric assessment-related factors between violent and non-violent offenders. Compared to non-violent offenders, violent offenders had a significantly higher prevalence of a history of violent crime (28.6% vs. 19.2%, $p = 0.005$).

**Table 2. Comparison of clinical factors between violent and non-violent offenders.**

| | Violent offenders (n = 304) | | Non-violent offenders (n = 344) | | $\chi^2$ | p value |
|---|---|---|---|---|---|---|
| | n | % | n | % | | |
| Family history | | | | | | |
| Severe mental illness | 38 | 12.5 | 50 | 14.5 | 0.569 | 0.491 |
| Schizophrenia spectrum and other psychotic disorders | 31 | 10.2 | 41 | 11.9 | 0.484 | 0.532 |
| Bipolar and other related disorder | 8 | 2.6 | 10 | 2.9 | 0.045 | 1.000 |
| Depressive disorder | 21 | 6.9 | 41 | 11.6 | 4.216 | 0.044 |
| Substance-related and addictive disorders | 37 | 12.2 | 52 | 15.1 | 1.182 | 0.304 |
| Intellectual disability | 14 | 4.6 | 16 | 4.7 | 0.001 | 1.000 |
| Suicide history | 35 | 11.5 | 34 | 9.9 | 0.450 | 0.525 |
| No prior psychiatric evaluation | 75 | 24.7 | 37 | 10.8 | 21.858 | <0.001 |
| Poor treatment adherence | 277 | 91.1 | 322 | 93.6 | 1.427 | 0.238 |

**Table 3. Comparison of forensic factors between violent and non-violent offenders.**

| | Violent offenders (n = 304) | | Non-violent offenders (n = 344) | | χ² | p value |
|---|---|---|---|---|---|---|
| | n | % | n | % | | |
| History of violent crimes | 87 | 28.6 | 66 | 19.2 | 7.961 | 0.005 |
| Forensic diagnosis | | | | | | |
| Severe mental illness | 129 | 42.4 | 162 | 47.1 | 1.416 | 0.237 |
| Schizophrenia spectrum and other psychotic disorders | 116 | 38.2 | 141 | 41.0 | 0.540 | 0.470 |
| Bipolar and other related disorder | 13 | 4.3 | 21 | 6.1 | 1.085 | 0.378 |
| Depressive disorder | 30 | 9.9 | 42 | 12.2 | 0.895 | 0.382 |
| Substance-related and addictive disorders | 92 | 30.3 | 92 | 26.7 | 0.983 | 0.338 |
| Neurocognitive disorders | 18 | 5.9 | 29 | 8.4 | 1.510 | 0.229 |
| Intellectual disability | 38 | 12.5 | 52 | 15.1 | 0.924 | 0.364 |
| Personality disorders | 9 | 3.0 | 4 | 1.2 | 2.653 | 0.159 |

### 3.4. Analysis of static and dynamic violent crime factors

Table 4 presents the associations between static and dynamic crime factors and violent offenders. Male (OR: 3.87; 95% CI: 2.42–6.19), absence of prior psychiatric evaluation before the offense (OR: 2.38; 95% CI: 1.50–3.78), and a history of violent crime (OR: 1.73; 95% CI: 1.10–2.72) were significantly associated with an increased likelihood of being a violent offender. Conversely, being single (OR: 0.34; 95% CI: 0.18–0.64) was associated with a reduced likelihood of being a violent offender. The model showed a good fit (Hosmer–Lemeshow statistics = 6.137, $p$ = 0.632).

### 3.5. Moderation analysis

Table 5 summarizes the results of the moderation analysis. Gender, treatment adherence, history of violent crime, and substance-related and addictive disorders did not significantly moderate the association between severe mental illness and violent crime.

**Table 4. The associations between static and dynamic crime factors and violent offenders.**

| | Violent offenders | |
|---|---|---|
| | OR | 95% CI |
| Static factors | | |
| Age | 0.99 | 0.98–1.01 |
| Male gender | 3.87 | 2.42–6.19 |
| Incomplete national education | 1.19 | 0.76–1.87 |
| Family history of severe mental illness | 0.90 | 0.56–1.48 |
| No prior psychiatric evaluation | 2.38 | 1.50–3.78 |
| History of violent crimes | 1.73 | 1.10–2.72 |
| Dynamic factors | | |
| Unemployed status | 0.75 | 0.43–1.32 |
| Single status | 0.34 | 0.18–0.64 |
| Severe mental illness | 1.11 | 0.78–1.59 |
| Substance-related and addictive disorders | 1.08 | 0.74–1.57 |
| Poor treatment adherence | 0.79 | 0.42–1.47 |
| Hosmer–Lemeshow goodness-of-fit test p value = 0.632 | | |
| OR, odds ratio; CI, confidence interval | | |

**Table 5. Moderation analysis.**

| | Coefficient | SE | Z | p value | 95% CI |
|---|---|---|---|---|---|
| Gender on the association between severe mental illness and violent crime | | | | | |
| Severe mental illness | −0.60 | 0.57 | −1.05 | 0.29 | −1.71–0.51 |
| Gender | 1.32 | 0.39 | 3.41 | <0.01 | 0.56–2.09 |
| Gender × Severe mental illness | 0.51 | 0.60 | 0.85 | 0.40 | −0.67–1.69 |
| Treatment adherence on the association between severe mental illness and violent crime | | | | | |
| Severe mental illness | −0.11 | 0.63 | −0.17 | 0.86 | −1.35–1.13 |
| Treatment adherence | −0.14 | 0.50 | −0.28 | 0.78 | −1.12–0.84 |
| Treatment adherence × Severe mental illness | −0.40 | 0.66 | −0.06 | 0.95 | −1.34–1.26 |
| History of violent crime on the association between severe mental illness and violent crime | | | | | |
| Severe mental illness | −0.14 | 0.21 | −0.66 | 0.51 | −0.56–0.28 |
| History of violent crime | 0.24 | 0.32 | 0.74 | 0.46 | −0.39–0.86 |
| History of violent crime × Severe mental illness | −0.41 | 0.49 | −0.08 | 0.93 | −1.00–0.92 |
| Substance-related and addictive disorders on the association between severe mental illness and violent crime | | | | | |
| Severe mental illness | −0.04 | 0.22 | −0.19 | 0.85 | −0.47–0.39 |
| Substance-related and addictive disorders | −0.38 | 0.25 | −1.52 | 0.13 | −0.87–0.11 |
| Substance-related and addictive disorders × Severe mental illness | −0.53 | 0.50 | −1.06 | 0.29 | −1.52–0.45 |

SE, standard error; CI, confidence interval

## 3.6. Sensitivity analysis

As part of the sensitivity analysis, we examined whether crime classification influenced the study results by conducting the same set of comparisons and analyses using the most violent and the most common non-violent crime categories—homicide and theft. Sociodemographic, clinical, and forensic factors were compared between homicide and theft offenders, followed by regression analyses of static and dynamic factors and moderation analyses. As shown in Table S4 in S1 File, being male (OR: 3.44; 95% CI: 1.04–11.38) and having never undergone psychiatric evaluation prior to the offense (OR: 3.19; 95% CI: 1.03–9.89) were associated with a higher likelihood of being a homicide offender.

As presented in Table S5 in S1 File, none of the moderation effects were statistically significant, including gender, treatment adherence, history of violent crime, and substance-related and addictive disorders in moderating the association between severe mental illness and homicide offending.

## 4. Discussion

In this study, we examined the association between severe mental illness, substance-related and addictive disorders, and violent crime in a sample of 648 offenders referred for forensic psychiatric evaluation by prosecutors or judges. Additionally, we investigated whether gender, treatment adherence, history of violent crime, and comorbid substance-related and addictive disorders moderated this association. The findings did not support the research hypotheses. Among the static factors, male gender, absence of prior psychiatric evaluation before the offense, and a history of violent crime were the strongest predictors of violent crime. Conversely, being single—a dynamic factor—was more closely associated with non-violent crime.

In the preceding section, we reviewed large-scale database studies from Western countries [9,13], which have suggested that individuals with serious mental illness may be at higher risk of involvement in violent crime compared to the general population. However, our study did not use a general population comparison group. Instead, we focused on differentiating violent and non-violent offenders within a forensic sample. Our analysis revealed that serious mental illness was not significantly associated with violent offending in this offender population. This finding suggests that, among individuals

who have already committed crimes, other risk factors beyond serious mental illness may play a more prominent role in distinguishing violent offenders from non-violent ones.

The average age, gender distribution, and crime type proportions in this study sample closely align with national crime statistics [34]. Among diagnoses from forensic psychiatric evaluations, schizophrenia spectrum and other psychotic disorders were the most prevalent (39.7%), followed by substance-related and addictive disorders (28.4%), intellectual disability (13.9%), and depression (11.1%). Epidemiological data indicate that schizophrenia affects approximately 0.33–0.75% of the global population [37,38], with a domestic prevalence of approximately 0.44% [39]. The global prevalence of substance-related and addictive disorders is estimated at 2.2% [14], while alcohol use disorders affect approximately 3% of the domestic population [40]. Psychotic disorders induced by other substances are estimated to affect at least 0.02–0.03% of individuals [39]. Additionally, intellectual disability affects approximately 0.4% of the domestic population [41], while depression affects approximately 1.2% of the population [42]. The stark contrast between these general prevalence rates and the diagnostic distribution in this study underscores the distinct psychiatric profile of the forensic population examined.

It is essential to note that the sample in this study consisted of offenders referred for forensic psychiatric evaluation by judicial authorities, meaning that all individuals were diagnosed with a mental disorder following evaluation. A significant proportion (49%) had severe mental illness, with schizophrenia spectrum and other psychotic disorders being the most prevalent (39.7%). Among both violent and non-violent offenders, schizophrenia spectrum and other psychotic disorders remained the most common diagnoses, each exceeding 30%. However, regression analysis revealed no significant association between severe mental illness and violent crime. Similarly, other psychiatric diagnoses, including depression and substance-related and addictive disorders, showed no significant association with violent crime—contrasting with findings from previous research. This discrepancy suggests that while mental illness may differentiate offenders from the general population, it may not reliably distinguish violent from non-violent offenders within the offender group.

In this study, 28.4% of offenders were diagnosed with substance-related and addictive disorders—a prevalence notably higher than in general epidemiological samples. This disparity likely reflects the distinct characteristics of the offender population examined.

Furthermore, only 2% of the sample underwent a forensic psychiatric diagnosis of personality disorder, a rate lower than the global prevalence of 7.8% [43] but higher than the domestic epidemiological estimate of 0.02% [39]. This discrepancy raises questions about whether personality disorders are underdiagnosed in forensic psychiatric evaluations of offenders within the domestic judicial system or if domestic diagnostic criteria are more conservative than those used internationally. Further research is needed to explore this issue.

The populations studied, including individuals with severe mental illness and substance use disorders, are epidemiologically more prevalent among males. Similarly, offender populations, particularly violent offenders, are predominantly male, as widely documented.

Biological research on violent crime among Chinese males suggests that variations in serotonin receptor polymorphisms may contribute to the higher proportion of male offenders [44,45]. From a neurodevelopmental and animal model perspective, the relationship between violent crime and neurotransmitters, such as dopamine and serotonin [46] may be influenced by maternal smoking during pregnancy [47]. Additionally, elevated testosterone levels have been proposed as a factor in male violent offenders [48]. Brain imaging studies further indicate that male violent offenders exhibit reduced gray matter volume in the insula compared to their non-violent counterparts [49], which may impair impulse control and emotional regulation. However, gender-based crime studies must account for potential biases. Higher arrest rates and harsher sentencing for male offenders than females could influence the study samples [50]. Understanding violent crime through a gendered lens may also benefit from sociological frameworks, such as differential association and social control theories, which examine the role of socialization and external influences on offender behavior. These perspectives provide a broader understanding beyond biological determinism, avoiding an oversimplified causal link between sex and violent crime [51].

Of the 648 individuals in this study, 112 had never undergone a psychiatric evaluation before the crimes. Over 50% had no history of violent crime. Following forensic psychiatric evaluation, 23 (20%) were diagnosed with severe mental illness, while 37 (30%) had substance-related and addictive disorders. Among these, 24 (64.9%) had alcohol use disorders, and 15 (40.5%) had stimulant use disorders. The remaining 52 (46.4%) were diagnosed with depression, neurocognitive disorders, intellectual disabilities, or other less common psychiatric conditions. These findings indicate that individuals classified as having "never received psychiatric evaluation before committing a crime" represent a highly heterogeneous group, with diagnoses extending beyond severe mental illness and substance-related and addictive disorders. This raises a critical question: Could the actual risk posed by individuals with untreated psychiatric conditions be systematically underestimated?

Regression analysis revealed a significant association between a history of violent crime and violent crime, whereas severe mental illness showed no such link. Similarly, moderation analysis revealed that gender, treatment adherence, history of violent crime, and substance-related and addictive disorders did not moderate the association between severe mental illness and violent crime. These findings indicate that offenders referred for forensic psychiatric evaluation due to violent crime are not necessarily those with severe mental illness.

Single status and unemployment are often cited as risk factors for criminal behavior. However, this study found no significant association between either factor and violent crime. Instead, single status was more closely associated with non-violent offenses. A large-scale Finnish study [52] similarly reported higher overall crime rates among single individuals, but the association between marital status and violent crime became insignificant when analyzed separately. These findings indicate that single status and unemployment may not directly increase the risk of crime but instead reflect underlying issues, such as inadequate social support or economic hardship. Further research is needed to clarify these relationships.

It is noteworthy that this study did not take into account the victims and situational factors involved in these cases. In future research, if such data can be adequately collected and incorporated into the analysis, it may allow for a more comprehensive exploration of how these factors influence the relationship between diagnosis and criminal behavior. In addition, we must also recognize the diversity of diagnoses, the blind spots in diagnostic descriptions, and the need to incorporate into future research the longitudinal dimension of symptom severity and impact within the diagnostic framework.

This study has several limitations. First, the classification and analysis of variables relied on forensic psychiatric evaluation reports, which limited the ability to comprehensively assess offenders' living environments, early life experiences, and objective measures of symptom severity. Second, inconsistencies in report quality were unavoidable. While some provided a detailed crime context, others differed in their descriptions of the relationship between circumstances and criminal behavior due to variations in evaluation objectives or focal points. Consequently, situational factors could not be quantitatively analyzed. Third, only 2% of the reports referenced a personality disorder diagnosis, and the terminology used for personality assessments varied, making it challenging to include personality traits and temperament as adjustment variables. Fourth, the sample primarily consisted of individuals referred by prosecutors for forensic psychiatric evaluation within a specific region, and most of the evaluations were conducted by a single institution. In the region where this study was conducted, Taiwan, negative stigmas toward mental illness still persist. Currently, there is no scientifically established standard for referring individuals for forensic psychiatric evaluation. Such referrals are typically made by prosecutors, and during the study period, defendants were not even legally guaranteed the right to request a psychiatric evaluation. Therefore, the representativeness of the sample is limited, and the generalizability of the findings should be interpreted with caution.

Finally, since the study sample consisted solely of offenders referred for forensic psychiatric evaluation, our findings have limited generalizability and cannot be directly compared to the prevalence or comorbidity patterns of mental disorders in the general population or among offenders who were not evaluated.

## 5. Conclusion

In this study, we examined offenders referred for forensic psychiatric evaluation and found that violent offenders who underwent such evaluations do not necessarily have severe mental illness or substance-related and addictive disorders. Moreover, factors such as male gender, poor treatment adherence, a history of violent crime, and comorbid addiction are insufficient to explain the relationship between severe mental illness and violent crime. It is crucial to avoid directly equating violent offenders with severe mental illness. Meanwhile, further research is needed to identify the factors that contribute to the involvement of individuals with severe mental illness in the criminal justice system. Efforts should also ensure they receive proper treatment and the necessary support for successful reintegration into society.

## Supporting information

**S1 File. Supplementary Tables.**
(DOCX)

**S2 Checklist. Inclusivity in global research questionnaire.**
(DOCX)

**S3 Data. Original dataset used for analyses.**
(XLSX)

## Author contributions

**Conceptualization:** Chia-heng Lin, Chia-Hsiang Chan.

**Data curation:** Chia-heng Lin, Wen-Ching Hsieh, Li-Ting Lin, Chia-Hsiang Chan.

**Formal analysis:** Chia-heng Lin, Li-Ting Lin, Chia-Hsiang Chan.

**Investigation:** Chia-heng Lin, Li-Ting Lin, Chia-Hsiang Chan.

**Methodology:** Chia-heng Lin, Wen-Ching Hsieh, Chia-Hsiang Chan.

**Software:** Chia-heng Lin.

**Supervision:** Chia-Hsiang Chan.

**Writing – original draft:** Chia-heng Lin.

**Writing – review & editing:** Chia-heng Lin.

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
