## [Decision Letter · Decision Letter 0]

17 Jul 2025

Factors associated with violent offenders with mental illness in forensic psychiatric evaluations

PLOS ONE

Dear Dr. Chan,

Thank you for submitting your manuscript to PLOS ONE. After careful consideration, we feel that it has merit but does not fully meet PLOS ONE’s publication criteria as it currently stands. Therefore, we invite you to submit a revised version of the manuscript that addresses the points raised during the review process.

Please, add your institutional e-mail to the article front page.

We look forward to receiving your revised manuscript.

Kind regards,

Vincenzo De Luca

Academic Editor

PLOS ONE

Journal Requirements:

“nil”

5. Please include your tables as part of your main manuscript and remove the individual files. Please note that supplementary tables (should remain/ be uploaded) as separate "supporting information" files

Reviewers' comments:

Reviewer's Responses to Questions

**Comments to the Author**

1. Is the manuscript technically sound, and do the data support the conclusions?

Reviewer #1: Yes

2. Has the statistical analysis been performed appropriately and rigorously?

Reviewer #1: Yes

3. Have the authors made all data underlying the findings in their manuscript fully available?

Reviewer #1: No

4. Is the manuscript presented in an intelligible fashion and written in standard English?

Reviewer #1: Yes

Reviewer #1: This retrospective study of Chinese forensic evaluation reports, N=648, tested the hypotheses: “(1) Severe mental illness, including schizophrenia and bipolar disorder, as well as substance-related and addictive disorders, are each associated with violent crime. (2) Gender, treatment adherence, history of violent crime, and comorbid substance-related and addictive disorders influence the relationship between severe mental illness and violent crime.”

They examined key demographic variables, including age, gender, education level, employment status, and marital status, plus clinical factors such as family history of mental illness, substance exposure history, suicide history, prior psychiatric evaluations, treatment adherence, and suicide history.

“Primary diagnostic categories found in the study included schizophrenia spectrum and other psychotic disorders, bipolar and related disorders, depressive disorders, substance-related and addictive disorders, neurocognitive disorders, personality disorders, and intellectual disabilities. For analytical purposes, schizophrenia spectrum and other psychotic disorders, along with bipolar and related disorders, are grouped under the severe mental illness category”.

“Bivariate analyses will be conducted using independent sample t-tests for continuous variables and chi-square tests for categorical variables. Multiple logistic regression will be performed to examine associations between static and dynamic crime factors (independent variables) and offender classification (violent vs. nonviolent) as the dependent variable.”

All of the above statistical design and results appear to be appropriate, but I was confused as to why all of the description in the Methods was written in future tense instead of past tense. Their conclusion is “These findings indicate that offenders referred for forensic psychiatric evaluation due to violent crime are not necessarily those with severe mental illness”.

Although they have some interesting neurochemical speculation on the reason for violent crime in their sample, I did not see an adequate rebuttal to the papers in their literature that DID find an association of major mental illness with violent crime.

Apart from that, their methods and conclusions appear to have been well-done.

**Do you want your identity to be public for this peer review?** For information about this choice, including consent withdrawal, please see our Privacy Policy

Reviewer #1: No

---

## [Author Response · Author response to Decision Letter 1]

5 Aug 2025

Dear Editors,

Thank you very much for your time and the opportunity to revise our manuscript entitled “Factors associated with violent offenders with mental illness in forensic psychiatric evaluations.” We greatly appreciate the constructive comments and suggestions provided by you and the reviewers, which have been very helpful in improving the quality of our manuscript.

We have carefully revised the manuscript in accordance with the reviewer’s comments. Below, we provide detailed responses to each point raised.

Additional Requirement:

Response: Thank you for the reminding. We have checked the format of out manuscript.

Response: We have completed the questionnaire and attached the information over the end part of the manuscript.

“nil”

Response: Sure. Thanks.

Response: Sure. We had uploaded the de-identified data onto the site The Qualitative Data Repository https://qdr.syr.edu/ The DOI is https://doi.org/10.5064/F6JZOYI6. Please also refer to the site https://data.qdr.syr.edu/dataset.xhtml?persistentId=doi%3A10.5064/F6JZOYI6

5. Please include your tables as part of your main manuscript and remove the individual files. Please note that supplementary tables (should remain/ be uploaded) as separate "supporting information" files

Response: Sure. We have included the tables into our manuscript.

Response: The reviewer did not mention some citations of published works.

Reviewer #1

1. All of the above statistical design and results appear to be appropriate, but I was confused as to why all of the description in the Methods was written in future tense instead of past tense.

Response: Thank you for the reminding. We have changed the future-tense descriptions in the method section to past-tense uses. Kindly refer to lines 220-234 on page 11.

“This study includes both continuous and categorical variables across demographic, clinical, and forensic evaluation-related factors. Continuous variables were reported as means and standard deviations, while categorical variables were expressed as frequencies and percentages. Bivariate analyses were conducted using independent sample t-tests for continuous variables and chi-square tests for categorical variables. Multiple logistic regression was performed to examine associations between static and dynamic crime factors (independent variables) and offender classification (violent vs. non-violent) as the dependent variable. Independent variables were entered into the statistical model using the Enter method. Model fit in logistic regression analysis were assessed using the Hosmer–Lemeshow goodness-of-fit test. Statistical significance was set at p < 0.05, with all tests conducted as two-tailed analyses.

To assess whether the static factors—male gender and history of violent crime—and the dynamic factors—poor treatment adherence and diagnoses from forensic psychiatric evaluation of substance-related and addictive disorders—moderate the relationship between severe mental illness and violent crime, a moderation analysis was conducted using Hayes’ PROCESS macro in SPSS [36]. Each factor was examined individually, with other factors included as covariates to control for potential confounding effects.”

2. Their conclusion is “These findings indicate that offenders referred for forensic psychiatric evaluation due to violent crime are not necessarily those with severe mental illness.” Although they have some interesting neurochemical speculation on the reason for violent crime in their sample, I did not see an adequate rebuttal to the papers in their literature that DID find an association of major mental illness with violent crime.

Response: Thank you for your kind reminder. To enhance the alignment with the reviewed literature, we have revised the relevant descriptions in the Discussion section. Kindly refer to lines 295-302 on page 17.

“In the preceding section, we reviewed large-scale database studies from Western countries [9,13], which have suggested that individuals with serious mental illness may be at higher risk of involvement in violent crime compared to the general population. However, our study did not use a general population comparison group. Instead, we focused on differentiating violent and non-violent offenders within a forensic sample. Our analysis revealed that serious mental illness was not significantly associated with violent offending in this offender population. This finding suggests that, among individuals who have already committed crimes, other risk factors beyond serious mental illness may play a more prominent role in distinguishing violent offenders from non-violent ones.”

---

## [Decision Letter · Decision Letter 1]

12 Oct 2025

Dear Dr. Chan,

Thank you for submitting your manuscript to PLOS ONE. After careful consideration, we feel that it has merit but does not fully meet PLOS ONE’s publication criteria as it currently stands. Therefore, we invite you to submit a revised version of the manuscript that addresses the points raised during the review process.

We look forward to receiving your revised manuscript.

Kind regards,

Vincenzo De Luca

Academic Editor

PLOS ONE

Journal Requirements:

Reviewers' comments:

Reviewer's Responses to Questions

**Comments to the Author**

Reviewer #1: All comments have been addressed

Reviewer #2: All comments have been addressed

Reviewer #3: (No Response)

2. Is the manuscript technically sound, and do the data support the conclusions?

Reviewer #1: Yes

Reviewer #2: Yes

Reviewer #3: Yes

3. Has the statistical analysis been performed appropriately and rigorously?

Reviewer #1: (No Response)

Reviewer #2: Yes

Reviewer #3: Yes

4. Have the authors made all data underlying the findings in their manuscript fully available?

Reviewer #1: Yes

Reviewer #2: (No Response)

Reviewer #3: (No Response)

5. Is the manuscript presented in an intelligible fashion and written in standard English?

Reviewer #1: Yes

Reviewer #2: Yes

Reviewer #3: Yes

Reviewer #1: Thank you for your revised manuscript. It answers all my previous questions and as it turns out, I do not have any further queries for you.

Reviewer #2: An interesting and easy to read paper that adds on to the literature on SUDs and concurrent disorders in an important way. Very minor writing edits needed (e.g., had a lower proportion of single "status" in line 242). Please describe the statement, "Compared to non-violent offenders, violent offenders had a significantly higher prevalence of a history of violent crime (28.6% vs. 19.2%, p = 0.005)" in line 263 and why the 19.2% of non-violent offenders with a history of violent crime would not be considered as violent offenders. Please also add on a big limitation of the sample being derived from a single site. May be fruitful to add in the Discussion the possibility of a combination of different factors over defined moderators that relates to violent crime.

Reviewer #3: This study addresses an important and sensitive topic—the association between severe mental illness and violent crime—using a large sample of forensic psychiatric evaluations in Taiwan. The work contributes valuable data from a non-Western context, but the paper would benefit from deeper theoretical integration and a clearer justification of analytic choices and interpretations.

Major Comments

– The Introduction would benefit from a clearer theoretical model linking mental illness, substance use, and violence. The current framing is largely descriptive and does not sufficiently articulate how this study advances existing literature or addresses prior inconsistencies.

– The manuscript states that all participants were referred for forensic evaluation, which introduces a selection bias. This limitation should be more explicitly acknowledged and discussed in relation to the generalizability of findings.

– Although based on established sources, the broad inclusion of offenses (e.g., intimidation, unlawful detention) may dilute distinctions between severe and moderate violence. The authors should justify this classification or provide sensitivity analyses using narrower definitions.

– The regression and moderation analyses appear correctly applied, but the rationale for including certain covariates is not always clear. It would strengthen the paper to clarify variable selection criteria and to report multicollinearity diagnostics.

– The conclusion that severe mental illness is not associated with violent crime within this sample is important, but the authors should avoid overgeneralizing this result. The Discussion could better explore alternative explanations, such as limited variance in psychiatric diagnoses or unmeasured mediating factors (e.g., symptom severity, treatment duration).

– Given that this study is based in Taiwan, the discussion could better highlight sociocultural aspects (e.g., mental health stigma, judicial referral patterns) that may influence both the prevalence of referrals and observed associations.

Minor Comments

- The tense in some sections (especially the Methods) has been corrected, but residual inconsistencies remain; please ensure uniform use of the past tense.

- The tables are informative but dense; consider summarizing key comparisons in the text rather than repeating all numerical results.

- Please verify reference formatting for PLOS ONE style (e.g., spacing and DOI presentation).

- Some typographical inconsistencies appear in the use of “addictive” vs. “additive” disorders.

- The Highlights section could be shortened to emphasize novel findings and clinical implications rather than restating results.

**Do you want your identity to be public for this peer review?** For information about this choice, including consent withdrawal, please see our Privacy Policy

Reviewer #1: No

Reviewer #2: No

Reviewer #3: No

---

## [Author Response · Author response to Decision Letter 2]

20 Oct 2025

Dear Editors,

We would like to express our sincere gratitude to the editorial team for their kind assistance and valuable guidance, which have greatly improved our manuscript. We hope that our responses this time will clearly convey our respect and appreciation to the reviewers. Our detailed replies are as follows.

Reviewer #2:

1. Very minor writing edits needed (e.g., had a lower proportion of single "status" in line 242).

Response: Thank you to the reviewer. We added the word status after single in line 261.

2. Please describe the statement, "Compared to non-violent offenders, violent offenders had a significantly higher prevalence of a history of violent crime (28.6% vs. 19.2%, p = 0.005)" in line 263 and why the 19.2% of non-violent offenders with a history of violent crime would not be considered as violent offenders.

Response: Thank you to the reviewer. To enhance clarity and avoid redundancy, we added a description in lines 186–187 of the Methods section, emphasizing that violent offenders were defined as individuals whose charges at the time of the forensic evaluation were categorized as violent crimes by the prosecutor.

3. Please also add on a big limitation of the sample being derived from a single site.

Response: Thank you to the reviewer. We added the following description in the Limitation section (lines 421–428):

“Fourth, the sample primarily consisted of individuals referred by prosecutors for forensic psychiatric evaluation within a specific region, and most of the evaluations were conducted by a single institution. In the region where this study was conducted, Taiwan, negative stigmas toward mental illness still persist. Currently, there is no scientifically established standard for referring individuals for forensic psychiatric evaluation. Such referrals are typically made by prosecutors, and during the study period, defendants were not even legally guaranteed the right to request a psychiatric evaluation. Therefore, the representativeness of the sample is limited, and the generalizability of the findings should be interpreted with caution.”

4. May be fruitful to add in the Discussion the possibility of a combination of different factors over defined moderators that relates to violent crime.

Response: Thank you to the reviewer. After considering this valuable suggestion, we added a new paragraph in lines 407–410:

“It is noteworthy that this study did not take into account the victims and situational factors involved in these cases. In future research, if such data can be adequately collected and incorporated into the analysis, it may allow for a more comprehensive exploration of how these factors influence the relationship between diagnosis and criminal behavior.”

Reviewer #3:

1. The Introduction would benefit from a clearer theoretical model linking mental illness, substance use, and violence. The current framing is largely descriptive and does not sufficiently articulate how this study advances existing literature or addresses prior inconsistencies.

Response: Thank you to the reviewer for this insightful comment. Currently, substance use is considered a potential interacting variable in the relationship between mental illness and violence, but existing studies vary in how much emphasis they place on this interaction. We believe that in our local context, where empirical research on this topic remains limited, it is still necessary to explore how to construct an integrated theoretical model. We added the following text in lines 145–148:

“We believe that through these data, we can identify variables potentially relevant to the relationship between mental illness, substance use, and criminal behavior. These variables warrant further investigation and may provide an opportunity to develop an integrated theoretical framework in this field.”

2. The manuscript states that all participants were referred for forensic evaluation, which introduces a selection bias. This limitation should be more explicitly acknowledged and discussed in relation to the generalizability of findings.

Response: Thank you to the reviewer. We added the following description in the Limitation section (lines 421–428):

“Fourth, the sample primarily consisted of individuals referred by prosecutors for forensic psychiatric evaluation within a specific region, and most of the evaluations were conducted by a single institution. In the region where this study was conducted, Taiwan, negative stigmas toward mental illness still persist. Currently, there is no scientifically established standard for referring individuals for forensic psychiatric evaluation. Such referrals are typically made by prosecutors, and during the study period, defendants were not even legally guaranteed the right to request a psychiatric evaluation. Therefore, the representativeness of the sample is limited, and the generalizability of the findings should be interpreted with caution.”

3. Although based on established sources, the broad inclusion of offenses (e.g., intimidation, unlawful detention) may dilute distinctions between severe and moderate violence. The authors should justify this classification or provide sensitivity analyses using narrower definitions.

Response: Thank you to the reviewer. To further clarify and distinguish this issue, we conducted an additional comparison between the most severe violent crime (homicide) and the most common non-violent crime (theft). We analyzed their sociodemographic, clinical, and forensic factors, as well as related interaction effects. These results were added to the Supplementary Materials. We also explained that this served as a sensitivity analysis to minimize the potential influence of crime classification on the findings. The results showed no substantial differences in the core associations. Please refer to lines 248–253:

“In addition, to minimize the potential influence of crime classification on the study results, we conducted the same set of comparisons and analyses using the most violent and the most common non-violent crime categories. Specifically, we compared homicide and theft offenders in terms of sociodemographic, clinical, and forensic factors, followed by regression analyses of static and dynamic factors, and finally, moderation analyses. This approach was intended to determine whether crime classification had a substantial impact on the study findings.”

And lines 305–315:

“As part of the sensitivity analysis, we examined whether crime classification influenced the study results by conducting the same set of comparisons and analyses using the most violent and the most common non-violent crime categories—homicide and theft. Sociodemographic, clinical, and forensic factors were compared between homicide and theft offenders, followed by regression analyses of static and dynamic factors and moderation analyses. As shown in Table S4, being male (OR: 3.44; 95% CI: 1.04–11.38) and having never undergone psychiatric evaluation prior to the offense (OR: 3.19; 95% CI: 1.03–9.89) were associated with a higher likelihood of being a homicide offender.

As presented in Table S5, none of the moderation effects were statistically significant, including gender, treatment adherence, history of violent crime, and substance-related and addictive disorders in moderating the association between severe mental illness and homicide offending.”

4. The regression and moderation analyses appear correctly applied, but the rationale for including certain covariates is not always clear. It would strengthen the paper to clarify variable selection criteria and to report multicollinearity diagnostics.

Response: Thank you to the reviewer. We cited relevant literature to support our inclusion of dynamic and static factors, ensuring that their selection was evidence-based. Additionally, we conducted a multicollinearity check using variance inflation factors (VIFs), which showed that all included variables had acceptable levels (VIF < 2.0), indicating no serious multicollinearity problems. We added the following description in lines 245–247:

“Multicollinearity diagnostics indicated acceptable levels across all predictors, with variance inflation factors (VIFs) below 2.0, suggesting that multicollinearity was not a concern in the regression models.”

5. The conclusion that severe mental illness is not associated with violent crime within this sample is important, but the authors should avoid overgeneralizing this result. The Discussion could better explore alternative explanations, such as limited variance in psychiatric diagnoses or unmeasured mediating factors (e.g., symptom severity, treatment duration).

Response: Thank you to the reviewer. We added the following discussion in lines 410-412:

“In addition, we must also recognize the diversity of diagnoses, the blind spots in diagnostic descriptions, and the need to incorporate into future research the longitudinal dimension of symptom severity and impact within the diagnostic framework.”

6. Given that this study is based in Taiwan, the discussion could better highlight sociocultural aspects (e.g., mental health stigma, judicial referral patterns) that may influence both the prevalence of referrals and observed associations.

Response: Thank you very much to the reviewer. We agree that these sociocultural factors influence the representativeness and generalizability of our findings. Therefore, we added the following description in the Limitation section (lines 423–428):

“In the region where this study was conducted, Taiwan, negative stigmas toward mental illness still persist. Currently, there is no scientifically established standard for referring individuals for forensic psychiatric evaluation. Such referrals are typically made by prosecutors, and during the study period, defendants were not even legally guaranteed the right to request a psychiatric evaluation.”

7. The tense in some sections (especially the Methods) has been corrected, but residual inconsistencies remain; please ensure uniform use of the past tense.

Response: Thank you to the reviewer. As we are not native English speakers, we had the manuscript reviewed by a native English editor. Upon discussion, we confirmed that variable definitions in the Methods section may appropriately use the present tense, while procedural descriptions should use the past tense. We carefully rechecked and corrected all remaining inconsistencies accordingly.

8. The tables are informative but dense; consider summarizing key comparisons in the text rather than repeating all numerical results.

Response: Thank you to the reviewer. We confirmed that the Results section now primarily highlights key comparative findings, with only a few representative descriptive statistics retained for context.

9. Please verify reference formatting for PLOS ONE style (e.g., spacing and DOI presentation).

Response: Thank you to the reviewer. We have rechecked, revised, and updated all references to comply with PLOS ONE formatting requirements.

10. Some typographical inconsistencies appear in the use of “addictive” vs. “additive” disorders.

Response: Thank you very much to the reviewer. We corrected all such typographical inconsistencies, primarily appearing in Tables 2–5.

11. The Highlights section could be shortened to emphasize novel findings and clinical implications rather than restating results.

Response: Thank you to the reviewer. We revised the Highlights as follows:

“�Stigmatization continues to hinder access to treatment, housing, and employment among individuals with mental illness.

Violent offending cannot be attributed solely to severe mental illness; diverse diagnostic and psychosocial factors are involved.

For offenders with mental illness, addressing clinical needs requires not only diagnostic evaluation but also integrated clinical and social interventions.

Social isolation and unmet support needs represent critical targets for preventing reoffending and facilitating recovery.”

---

## [Decision Letter · Decision Letter 2]

2 Nov 2025

Factors associated with violent offenders with mental illness in forensic psychiatric evaluations

PONE-D-25-24024R2

Dear Dr. Chan,

We’re pleased to inform you that your manuscript has been judged scientifically suitable for publication and will be formally accepted for publication once it meets all outstanding technical requirements.

Kind regards,

Vincenzo De Luca

Academic Editor

PLOS ONE

Additional Editor Comments (optional):

Reviewers' comments:

Reviewer's Responses to Questions

**Comments to the Author**

Reviewer #3: All comments have been addressed

2. Is the manuscript technically sound, and do the data support the conclusions?

Reviewer #3: Yes

3. Has the statistical analysis been performed appropriately and rigorously?

Reviewer #3: Yes

4. Have the authors made all data underlying the findings in their manuscript fully available?

Reviewer #3: Yes

5. Is the manuscript presented in an intelligible fashion and written in standard English?

Reviewer #3: Yes

Reviewer #3: I have no further comments. The paper is suitable to be accepted for publication in my opinion.

**Do you want your identity to be public for this peer review?** For information about this choice, including consent withdrawal, please see our Privacy Policy

Reviewer #3: No

---

## [Editor Report · Acceptance letter]

PONE-D-25-24024R2

PLOS ONE

Dear Dr. Chan,

I'm pleased to inform you that your manuscript has been deemed suitable for publication in PLOS ONE. Congratulations! Your manuscript is now being handed over to our production team.

Kind regards,

on behalf of

Dr. Vincenzo De Luca

Academic Editor

PLOS ONE